# Vegetation trends associated with urban development: The role of golf courses

**Thu Thi Nguyen** [1,2]*, **Paul Barber** [2,3], **Richard Harper** [2], **Tran Vu Khanh Linh** [4], **Bernard Dell** [2]

1 Vietnam National University of Forestry (VNUF), Ha Noi, Vietnam, 2 Agricultural and Forestry Sciences, Murdoch University, Murdoch, WA, Australia, 3 ArborCarbon Pty Ltd, East Perth, WA, Australia, 4 Faculty of Forestry, Nong Lam University—Ho Chi Minh City, Ho Chi Minh City, Vietnam

* Thu.Nguyen@murdoch.edu.au, thu.nguyen.2k14@gmail.com

**Data Availability Statement:** All relevant data are within the manuscript

**Funding:** Ms Thu Nguyen received scholarships from the Vietnam International Education Department (VIED) and Murdoch University to

## Abstract

Globally, cities are growing rapidly in size and density and this has caused profound impacts on urban forest ecosystems. Urbanization requiring deforestation reduces ecosystem services that benefit both city dwellers and biodiversity. Understanding spatial and temporal patterns of vegetation changes associated with urbanization is thus a vital component of future sustainable urban development. We used Landsat time series data for three decades from 1988 to 2018 to characterize changes in vegetation cover and habitat connectivity in the Perth Metropolitan Area, in a rapidly urbanising Australian biodiversity hotspot, as a case study to understand the impacts of urbanization on urban forests. Moreover, as golf courses are a major component in urban areas, we assessed the role of golf courses in maintaining vegetation cover and creating habitat connectivity. To do this we employed (1) land use classification with post-classification change detection, and (2) Morphological Spatial Pattern Analysis (MSPA). Over 17,000 ha of vegetation were cleared and the area of vegetation contributing to biodiversity connectivity was reduced significantly over the three decades. The spatial patterns of vegetation loss and gain were different in each of the three decades (1988–2018) reflecting the implementation of urban planning. Furthermore, MSPA analysis showed that the reduction in vegetation cover led to habitat fragmentation with a significant decrease in the core and bridge classes and an increase in isolated patches in the urban landscape. Golf courses played a useful role in maintaining vegetation cover and contributing to connectivity in a regional biodiversity hotspot. Our findings suggest that for future urban expansion, urban planning needs to more carefully consider the impacts of deforestation on connectivity in the landscape. Moreover, there is a need to take into consideration opportunities for off-reserve conservation in smaller habitat fragments such as in golf courses in sustainable urban management.

## 1 Introduction

Globally, cities have grown rapidly in number and size over recent decades [1, 2]. This trend is predicted to continue as urban areas are expected to absorb most global population growth

pursue her PhD degree at Murdoch University where this study was carried out. VIED did not play a role in the study design, data collection and analysis, decision to publish, or preparation of the manuscript. Murdoch University provided support to Professors Bernard Dell and Richard Harper in the form of salaries and provided support to Ms. Thu Nguyen in the form of facilities, but did not play a role in the study design, data collection and analysis, decision to publish, or preparation of the manuscript. ArborCarbon provided financial support for A/Professor Paul Barber, but did not have any additional role in the study design, data collection and analysis, decision to publish, or preparation of the manuscript. The specific roles of these authors are articulated in the 'author contributions' section.

**Competing interests:** A/Prof. Paul Barber is employed by the commercial entity ArborCarbon who provides in-kind support only through the employment of the co-author A/Prof. Paul Barber and access to facilities, equipment and software. This commercial affiliation does not alter our adherence to all PLOS ONE policies in sharing data and materials. Prof. Dell, Prof. Harper and other authors have no competing interests.

[3]. While the process of urbanization presents key implications for changes in physical landscapes and demographic characteristics, it can cause profound impacts on environmental components, especially on urban forest ecosystems [4, 5].

Vegetation in urban landscapes is critically important because provides goods and services, and full ecosystem functions that benefit city dwellers and the environment. On the one hand, a remarkable range of human well-being benefits are derived from urban green spaces including mitigating the urban heat island (UHI) effect which is a threat to human health [6, 7]; reducing stress [8], improving healing times [9], increasing self-esteem and empowerment [10], and improving cognitive ability [11]. On the other hand, urban green spaces provide various ecosystem services such as strengthening resistance to some kinds of natural disasters for example, floods [12], promoting biological processes such as pollination [13], and reducing surface erosion from stormwater runoff [14]. The amount of vegetation in cities strongly influences biodiversity, especially where vegetation is set aside during the process of urbanization [15]. However, if species dispersal and exchange among these patches is insufficient to allow gene flow and diversity, loss of regional biodiversity is inevitable [16]. Therefore, urban development that requires deforestation, with habitat loss and fragmentation is a threat to biodiversity [15, 17].

Urban conservation strategies must therefore consider not only the size and quality of habitat reserves, but the connectivity in the intervening urban vegetation matrix [17]. While the need to protect large habitat reserves is obvious, opportunities for off-reserve conservation of smaller habitat fragments should not be overlooked [18]. Understanding spatial and temporal patterns of vegetation change associated with urbanization, as well as opportunities for the preservation of green spaces outside natural reserves, is vital for future sustainable urban development especially in areas of global ecological importance.

As the number of golf courses is rapidly increasing in many urban areas worldwide [19], there have been many environmental arguments about this green space category in urban landscapes. Golf courses are sometime considered to be major polluters of the environment through pesticide and fertilizer use [20]. In fact, golf courses have been established for recreational purposes, which are a mix of bushland, fairways and infrastructure. Though they are not fully ecologically functional as with a natural parks, previous studies investigated the condition of vegetation inside the golf courses and indicated that the bushland in non-playing areas of golf courses are significantly important to biodiversity conservation and the provision of ecosystem services in cities [21], such as providing refuge habitats for urban-avoiding wildlife [22, 23, 24]. Therefore, potentially, together with the natural reserves, golf courses can play some roles as off-reserve sites for purposeful biodiversity conservation in urban landscapes. Nevertheless, little research has been undertaken to comprehensively assess the role of golf courses in maintaining vegetation patches as interconnected nodes in urban landscapes during a long period of urban development where deforestation was significant.

Deforestation for urban expansion can occur gradually over multiple years. Satellite-based remote sensing holds certain advantages in the characterization of these changes in urban landscapes because of the large spatial coverage, high time resolution, and wide availability of data [25]. Many methods have been used to detect, monitor and quantify vegetation changes, but differences in vegetation index and land use classification are the most widely used methods for vegetation changes over a long period of time [26].

A great number of vegetation indices have been proposed, ranging from very simple to very complex band combinations [27]. The most widely-used vegetation index is the Normalized Difference Vegetation Index (NDVI), which is an efficient and simple metric to identify vegetated areas and their condition [28]. This separates green vegetation from other surfaces based on the differential absorption of red light by chlorophyll and reflection of NIR by green

vegetation [29]. Furthermore, information on vegetation cover dynamics can be combined with Morphological Spatial Pattern Analysis (MSPA) to describe the spatial configuration of the ecosystem at the pixel level, making it possible to detect temporal changes in the structural connectivity of habitats in urban settings [30].

Therefore, to enhance the understanding of vegetation dynamics associated with urbanization and the role of golf courses in maintaining urban forests, we used Landsat imagery to map vegetation cover and to assess its spatial and temporal distribution over three decades from 1988 to 2018. Maps of vegetation cover were used for MSPA analysis to detect changes in habitat connectivity. We chose the Perth Metropolitan area for the study as it lies within a rapidly urbanizing biodiversity hotspot in Australia. The three primary research objectives were to: (1) determine the spatial and temporal patterns of deforestation in urbanization; (2) evaluate the spatial and temporal patterns of green landscape connectivity; and (3) assess the role of golf courses in preserving green spaces and biodiversity in an urban landscape. This analysis will provide a useful perspective on the land-use pressure facing vegetation remnants in this region which is recognised as one of 35 international biodiversity hotspots with over 1500 plant species with a high degree of endemism [31], and provide a framework for planning urban expansions both in this region and globally.

## 2 Methods

### 2.1 Study area

The study area belongs to the Perth Metropolitan region covering four sub-regions (North West, Middle Central, Inner Central and South West). Perth has a Mediterranean-type temperate climate with a hot and dry summer, and a cold and rainy season occurring between May and October [32]. Under future climate-change scenarios, this area is projected to experience a lower annual rainfall [33].

Perth belongs to the Australia's southwest corner, which is recognized as a global biodiversity "hotspot" with outstanding natural environments. Our study area occurs on the Swan Coastal Plain which is part of the southwest of Australia which in turn has the highest concentration of rare and endangered species on the entire continent (at least 1,500 endemic species). More than 6,000 species of native plants and 100 native mammals, birds, frogs and reptiles occur in this region, making it a biodiversity "hotspot" [31, 34].

Perth has retained a significant area of natural vegetation, which has conservation significance thanks to the introduction of legislation and policies aimed primarily at protecting biophysical environmental values. This includes the Western Australian Environmental Protection Act 1986 [35]; the Federal Government's Environment Protection and Biodiversity Conservation Act 1999 [36]; and more recently, Bush Forever [37, 38], a policy which took a whole-of-government approach to identify and protect biologically significant bushland and wetlands within the Perth metropolitan area. However, Perth has experienced extensive urban development since the 1980s [39]. The expansive growth of the Perth metropolitan footprint has contributed to the loss of biodiversity, together with the ecosystem services provided by natural areas. Together with urbanization, the golf industry has expanded contributing to a growing proportion of Perth's urban green space with 34 golf courses in the study area (Fig 1).

### 2.2 Approaches

We conducted two types of analyses in this study: (1) assessment of vegetation cover change in golf courses since 1988 relative to surrounding areas to provide a broad overall context of the urban vegetation changes that have taken place throughout the region and the role of golf courses; and (2) assessment of MSPA. In these analyses, we used four data sets (1988, 1998,

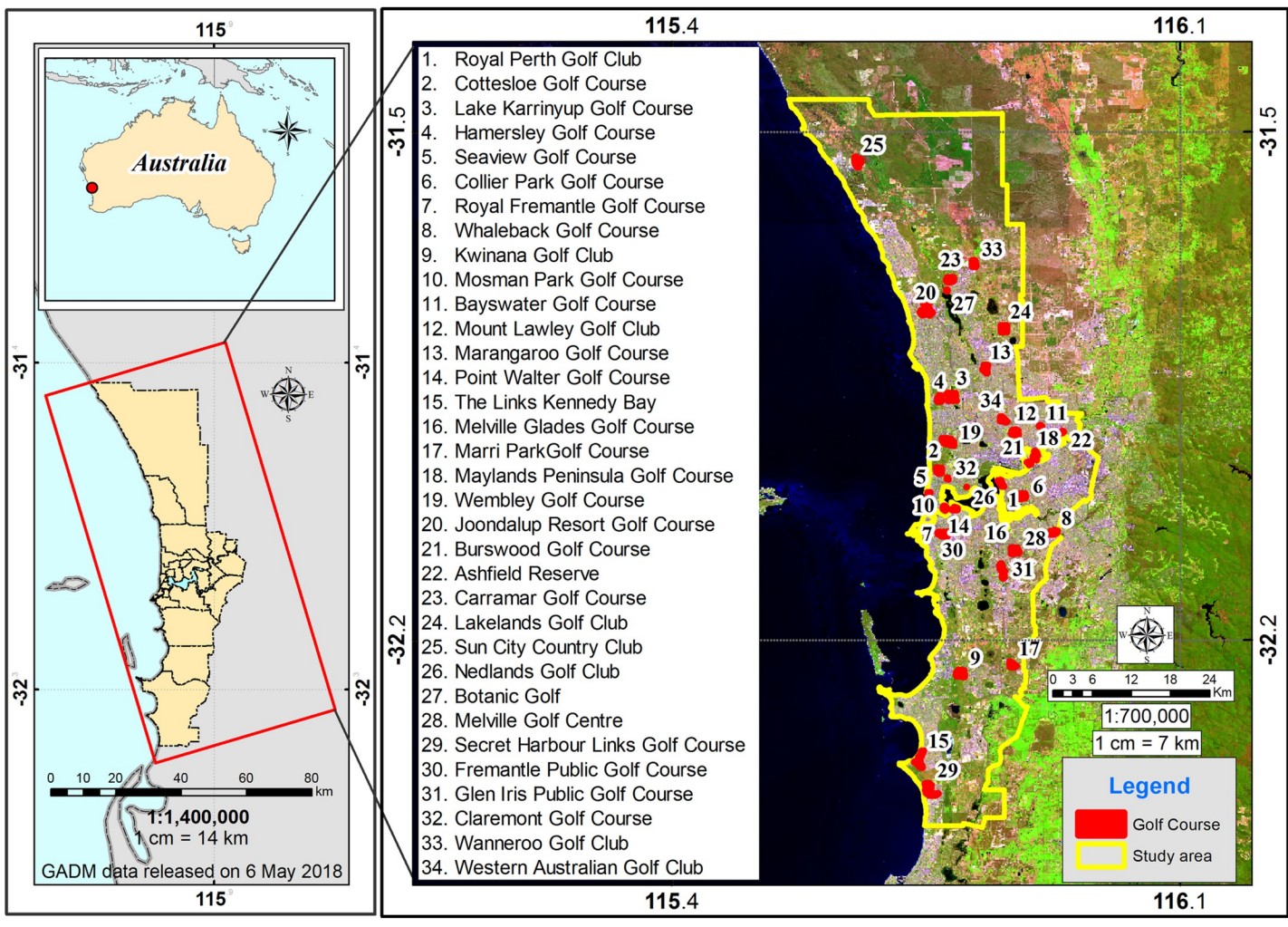

**Fig 1. Location of the study area in Perth, south-western Australia.** The satellite image was obtained from Landsat 8 on 30 September 2018 from the public domain: http://eros.usgs.gov of the Earth Resources Observatory and Science (EROS) Center. Location of golf courses are shown.

2008, 2018) covering thirty years of urban development. The steps taken in this study are summarized in a flow chart (Fig 2).

**2.2.1 Landsat data and pre-processing.** Three Landsat 5 Thematic Mapper images (WRS path 113 row 82) were acquired from 1988 to 2008 and one Landsat OLI 8 (WRS path 113 row 82) was acquired in 2018 at four time steps, including year 1988 (11th December), year 1998 (7th December), year 2008 (18th December), year 2018 (14th December). The Landsat imagery were obtained from the US Geological Survey (USGS) of the Earth Resources Observation and Science Center (EROS). Image dates were selected acquired during December (summer, dry season) to reduce the seasonal difference effects. All images selected are cloud-free scenes or little cloud cover (0.05% and 0.4%) scenes with the whole study area is cloud-free; Therefore, is no requirement for removing cloud. Georeferencing was performed at the USGS prior to downloading the data (L1T level of systematic geometric accuracy) and no further refinement was undertaken. Atmospheric and topographic corrections were performed on the Landsat data sets. The atmospheric correction was carried out to adjust the multitemporal dataset to a common radiometric scale [40].

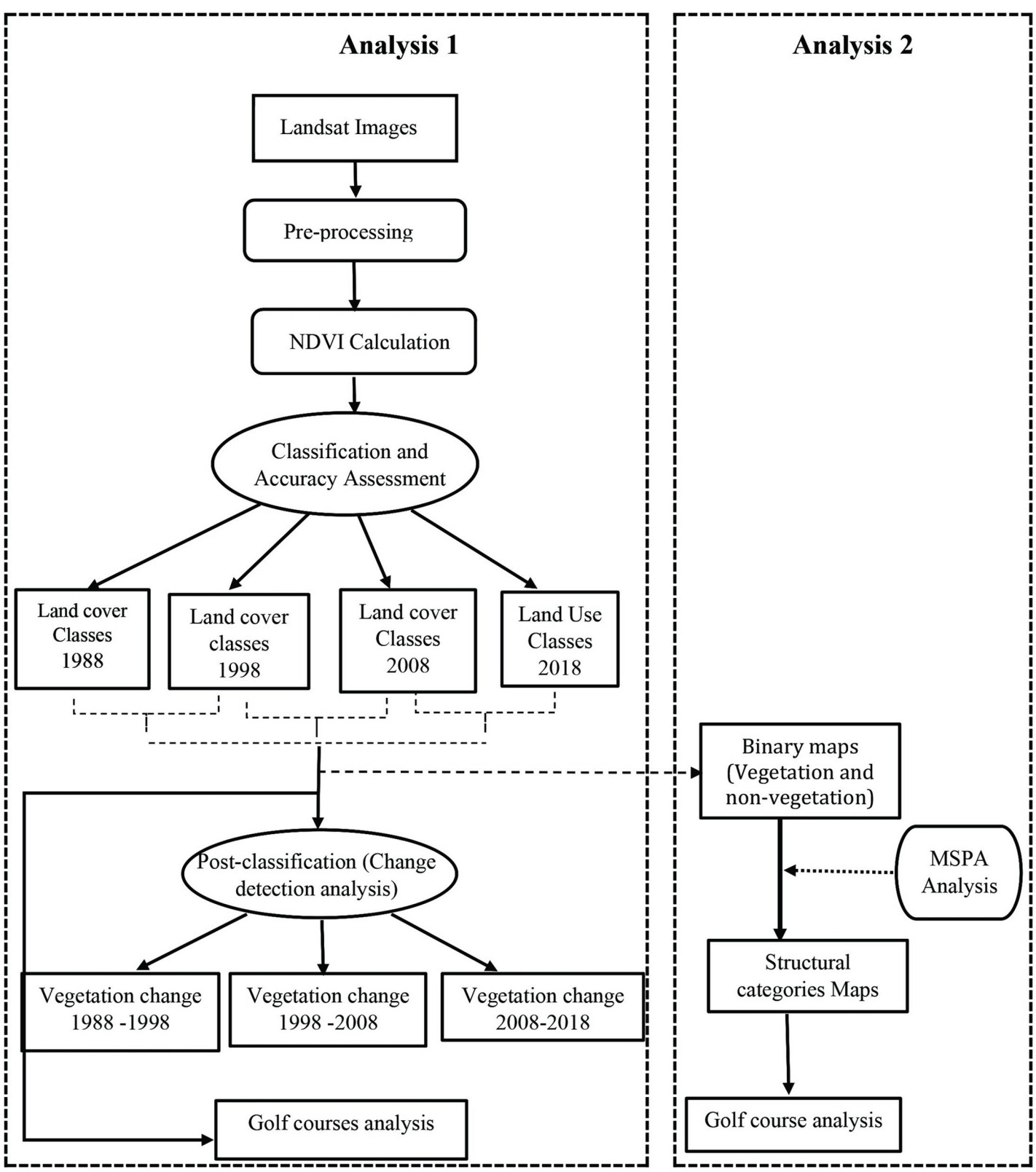

**Fig 2. Flow chart summarizing the major steps taken during the investigation.**

The first process of atmospheric correction was conversion of the digital number (DN) remote sensing data values to at-sensor radiance based on the image header file. After that we employed the image-based models—dark object subtraction (DOS) to correct atmospheric scattering scene-by-scene. This method is a widely used and effective method in atmospheric correction [41–47]. Topographic correction was conducted to remove topographic effects. We used a sun-canopy-sensor (SCS) correction based on the 30 m digital elevation model (DEM) because topographic shading is not only due to slope but also to shadowing of one tree crown over another and this is one of the most widely and effective used methods of topographic correction [48].

**2.2.2 Classification.**    In a preliminary step, we used a decorrelation stretch to enhance the image for more effective visualization. Prior to image classification, NDVI images were generated. A classification technique was then applied to the NDVI images of 1988, 1998, 2008 and 2018 using Arc-GIS 10.3 software. NDVI images were obtained by calculating the ratio between the red (R) and near infrared (NIR) values of the satellite image using Eq 1:

$$NDVI = \frac{(NIR - R)}{(NIR + R)} \tag{1}$$

In Landsat 4–7, NDVI = (Band 4 –Band 3) / (Band 4 + Band 3).
In Landsat 8, NDVI = (Band 5 –Band 4) / (Band 5 + Band 4).

Landsat TM data from different dates were independently classified based on the NDVI values. Water bodies have negative NDVI values, whereas, bare soil and built-up areas have an NDVI value of around zero. Chlorophyll in green vegetation, on the other hand, absorbs RED to drive photosynthesis thereby providing moderate and high NDVI values close to +1 [49]. Based on this understanding, the four NDVI images were classified into three classes (Vegetation, Built up + bare soil, Water bodies) using the NDVI threshold ranges technique in Arc-GIS 10.3 software.

The classification based on NDVI threshold was evaluated using accuracy assessment. An error matrix compared information from a classified image or land cover map to known reference (truth) sites for a number of sample points assessed in 2018. We obtained photographs of representative land use categories with GPS locations to assist in our image interpretations. Also, we used Google Earth images, true and false colour combination images and knowledge-based information including expert knowledge, land use maps and reports. For historical images, Google Earth was used to substitute the traditional reference data collection on each of the sites [50]. Based on data of accuracy assessment, we reclassified the preliminary land use classification maps to improve the accuracy of classification.

**2.2.3 Vegetation change detection.**    In order to detect the vegetation cover change, we created binary maps of vegetation and non-vegetation from the classified maps in the previous analysis, one for each adjacent pair of time steps, which depict where degradation occurred within a decade of urbanization. This post-classification analysis uses two images from different dates and classifies them independently. We then calculated changes in vegetation cover type using Eq 2:

$$Change\ area = D_2 - D_1 \tag{2}$$

where D1 and D2 are the area of the target vegetation cover at the beginning and the end of the study period, respectively. This analysis allows the calculation of vegetation loss and gain in each period.

**2.2.4 Golf courses analysis 1.**    We compared the vegetation cover, and the change (vegetation loss and gain) taking place within all golf courses in the study area, and in their

surrounding regions. After creating the GIS boundaries of the golf courses, we extracted the vegetation cover and the vegetation change within these boundaries at four time steps in 1988, 1998, 2008 and 2018 to compare vegetation dynamics within the golf courses and the whole study area over time.

**2.2.5 Morphological Spatial Pattern Analysis (MSPA) for the structural connectivity of habitats.** MSPA was employed to describe the structural connectivity of habitats in Perth for four time steps in 1988, 1998, 2008 and 2018. This method describes the spatial and temporal configuration of the ecosystem at the pixel level [30], which was based on the concept of "habitat availability" and "graphic theory" [51, 52] in which the landscape is considered as a collection of nodes, and links with a node is a place where connectivity exists and will depend on the width of itself. The output of the MSPA analysis includes the seven structural categories into which habitats are divided, including core, edge, perforation, bridge, loop, branch and islet [53] and [54] and is summarized in Table 1.

In order to undertake the MSPA analysis, we defined the input data (foreground class). For this study, we used the classified maps for 1988, 1998, 2008, 2018 in Analysis 1 to create the binary maps which contained vegetation and non-vegetation classes. Hence, the high and full covered vegetation pixels were defined as the foreground pixels (green landscape) in the MSPA approach. The results of MSPA analysis for the four time steps allowed us to assess the changes in habitat connectivity through time associated with urbanization.

**2.2.6 Golf course analysis 2.** To assess the role of golf courses in maintaining biodiversity connectivity over 30 years, we compared the habitat connectivity within all golf courses in the study area and in their surrounding green spaces. Using the GIS boundaries of the golf courses, we extracted the habitat connectivity within the golf course boundaries for four time steps (1988, 1998, 2008 and 2018).

# 3 Results

## 3.1 Land use classification in Perth

Data sets representing four time periods (1988, 1998, 2008 and 2018) are shown in Fig 3, Fig 4 and Table 3, which provide an overview of the land cover changes (vegetation, built up and

**Table 1. Definition of Morphological Spatial Pattern Analysis (MSPA) classes.**

| Class | Description | Ecological Meaning |
|---|---|---|
| Core | A collection of foreground pixels which are interconnected and greater than the user-specified edge width from background | Large-scale natural patches with high connectivity |
| Edge | Transition pixels between the foreground and background that form the outer edge | The transition zone between vegetation and non-vegetation areas. |
| Perforation | The transition pixels between foreground and background inside core areas that form the inner edge | Unnatural patch inside the core area. |
| Bridge | A set of linear foreground pixels between two cores that connect two or more core areas. | The striped ecological land that connects two cores, which is equivalent to the connecting corridor of the green space network. |
| Loop | Linearly oriented foreground pixels extended from core that connects core area to itself. | Connecting corridor inside a large natural patch. |
| Branch | Linearly oriented foreground pixels extended from core that do not connect to any other core area | Striped ecological land with low connectivity. |
| Islet | A collection of foreground pixels which is smaller than the core zone and do not connect to any other foreground cells. | Small natural patches that are isolated and do not connect to each other. |

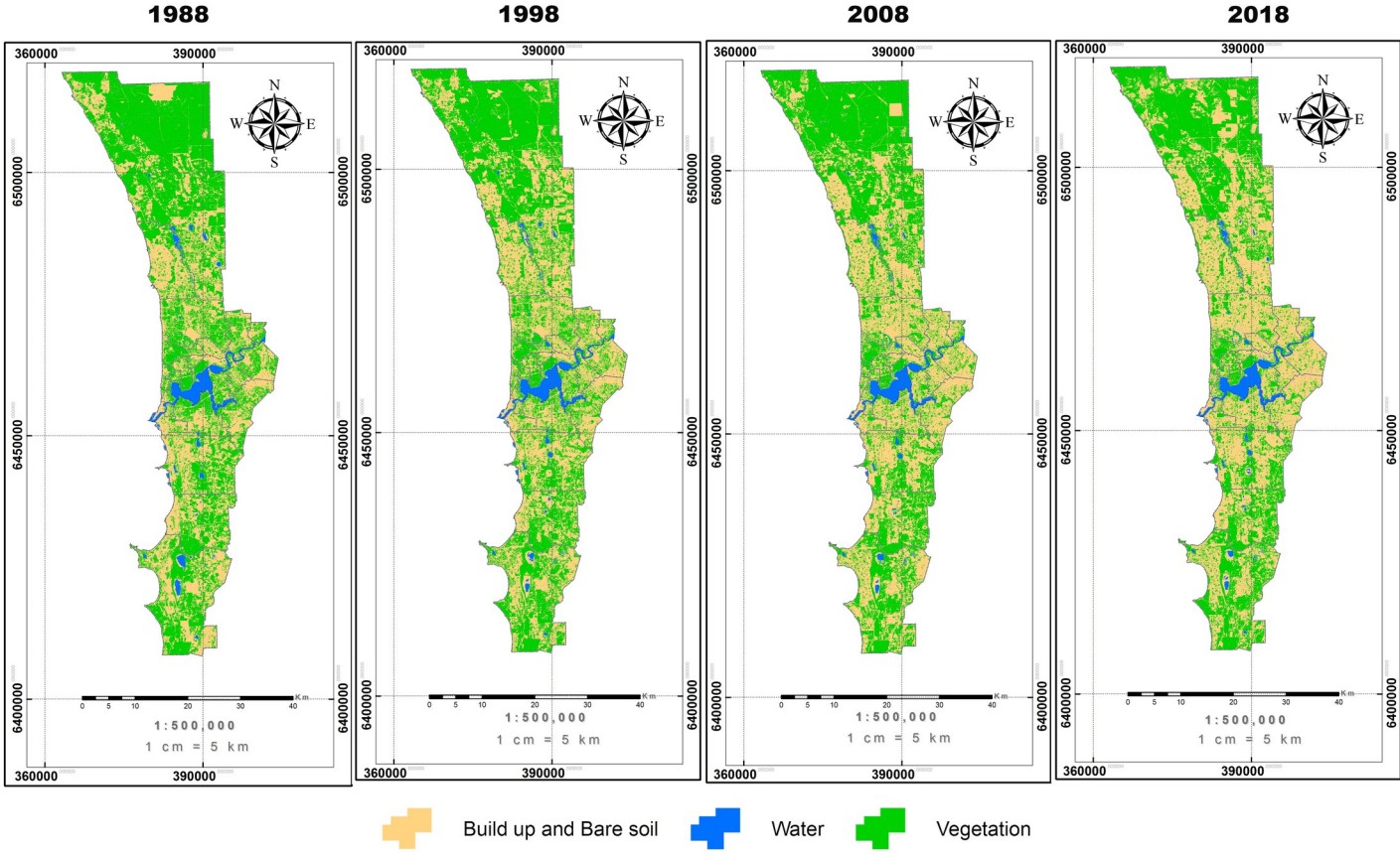

**Fig 3. Map of land cover classification for Perth in four time steps between 1988 and 2018.** The three classes of land cover shown are vegetation, water, and build up and bare soil.

bare land, water bodies) over recent decades. From the 1988 and 1998 data sets, it is evident that over half of the region was vegetated. However, the urban footprint of built up and bare land area had increased 10% from 1998 to 2018. As a consequence, there was a significant decrease in vegetation cover, which comprised 56% of the land surface in 1988 and declined by 10.1% over the next 30 years (Table 2).

The analysis had the overall accuracy (OA) of classification from 87% and the kappa coefficient from 91% for the three classes (Table 3).

Using the GIS layer of land use categories, we also extracted the data for three classes within the golf courses in comparison with the whole area (Table 2). Our analysis shows that while there was a significant decrease in vegetation cover throughout the region, the total area of golf courses remained unchanged at around 1,093 ha over the last 30 years.

## 3.2 Spatial patterns of vegetation change

To characterize spatial patterns of vegetation dynamics, we detected the vegetation loss and gain for each of the three decades (Fig 5). In the period 1988 to 1998, deforestation occurred intensively in the central and north regions. From 1998 to 2008, vegetation loss expanded to the north and south of the city. However, in the last decade from 2008 to 2018, vegetation loss accelerated in the distal regions with urbanization. However, there was also some vegetation gain over the three decades (Fig 5, Table 4) predominantly in the northern part of the city.

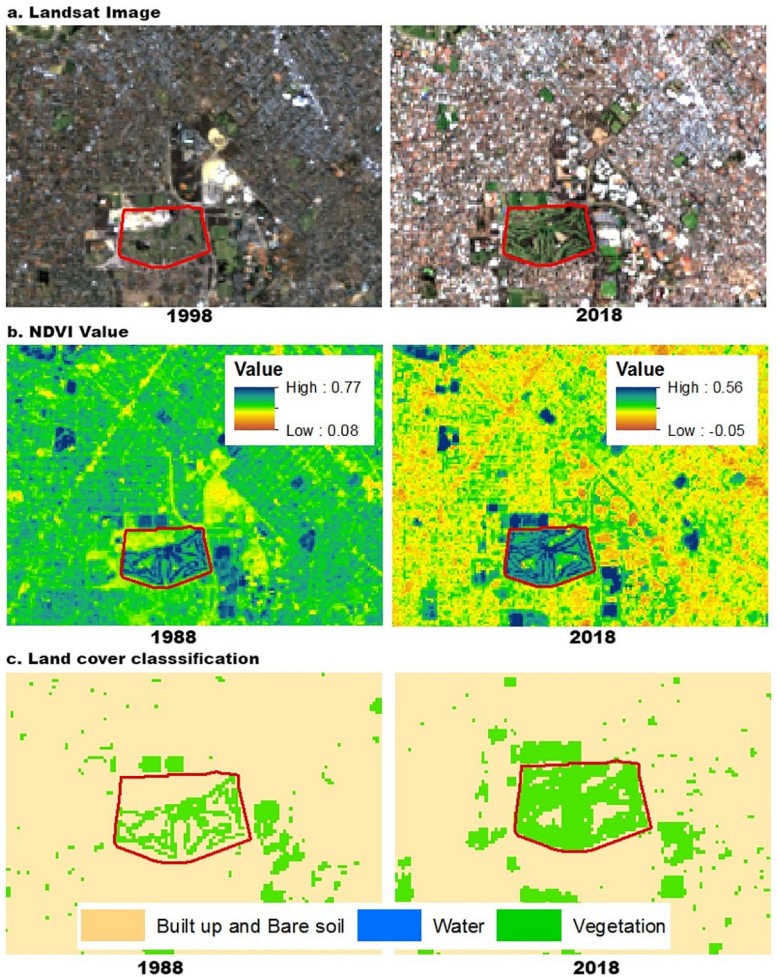

**Fig 4.** Land cover classification showing a detailed view of the Collier Park Golf Course for Landsat image (a), NDVI values (b),and Land cover classification (c).

Calculation of changes within the golf courses and in the whole area (Table 4) showed that the major urban area of Perth experienced a net loss in vegetation cover. Though vegetation compensation occurred together with deforestation over 30 years of urbanization, the vegetation loss was always much larger than vegetation gain with the largest net vegetation loss occurring in the last decade. However, the golf courses showed a different trend where the net gain of vegetation cover happened over three decades and the largest gain was between 1988 and 1998.

### 3.3 Analysis of connectivity components of green space networks

Results of the MSPA analysis indicate that the reduction in vegetation cover over the last thirty years has led to a decline in connectivity (Table 5). Among the two MSPA classes that are important for connectivity (core and bridge), the total area of core class decreased by about 10% over three decades while the bridge class was maintained at around 37,000 ha, but the proportion of this class per total vegetation cover area (VCA) increased due to the reduction of vegetation cover over time (Table 5).This analysis also shows the fluctuation in the areas of the rest of the MSPA classes including islets, loops, edges, perforations, and branches which do

**Table 2. Land cover classification within the Perth Metropolitan Region.**

| | | Land cover category | | | | | |
|---|---|---|---|---|---|---|---|
| | | Whole area | | | Golf course | | |
| | | Vegetation | Built up and bare soil | Water bodies | Vegetation | Built up and bare soil | Water bodies |
| 1988 | Area (ha) | 98,446 | 74,376 | 2,667 | 929 | 210 | 0.1 |
| | Proportion (%) | 56.1 | 42.4 | 1.5 | 81.5 | 18.5 | 0.0 |
| 1998 | Area (ha) | 91,754 | 81,464 | 2,271 | 1,042 | 97 | 0.7 |
| | Proportion (%) | 52.3 | 46.4 | 1.3 | 91.4 | 8.5 | 0.1 |
| 2008 | Area (ha) | 88,341 | 84,971 | 2,176 | 1,084 | 55 | 0.2 |
| | Proportion (%) | 50.3 | 48.4 | 1.2 | 95.1 | 4.9 | 0.0 |
| 2018 | Area (ha) | 80,755 | 92,243 | 2,491 | 1,093 | 46 | 0.7 |
| | Proportion (%) | 46.0 | 52.6 | 1.4 | 95.9 | 4 | 0.1 |

**Table 3. Accuracy assessment of land cover maps generated.**

| Year 1988 | | | | |
|---|---|---|---|---|
| LULC Class | Vegetation | Build Up and Bare Soil | Water Bodies | Total |
| Vegetation | 14 | 1 | 0 | 15 |
| Build Up and Bare Soil | 1 | 12 | 2 | 15 |
| Water Bodies | 0 | 0 | 15 | 15 |
| Total | 15 | 13 | 17 | 45 |
| | | Overal Accuracy | 87% | |
| | | Overal Kappa | 0.91 | |
| **Year 1998** | | | | |
| LULC Class | Vegetation | Build Up and Bare Soil | Water Bodies | Total |
| Vegetation | 15 | 0 | 0 | 15 |
| Build Up and Bare Soil | 0 | 13 | 2 | 15 |
| Water Bodies | 1 | 0 | 14 | 15 |
| Total | 16 | 13 | 16 | 45 |
| | | Overal Accuracy | 90% | |
| | | Overal Kappa | 0.93 | |
| **Year 2008** | | | | |
| LULC Class | Vegetation | Build Up and Bare Soil | Water Bodies | Total |
| Vegetation | 14 | 1 | 0 | 15 |
| Build Up and Bare Soil | 1 | 14 | 0 | 15 |
| Water Bodies | 0 | 0 | 15 | 15 |
| Total | 15 | 15 | 15 | 45 |
| | | Overal Accuracy | 93% | |
| | | Overal Kappa | 0.96 | |
| **Year 2018** | | | | |
| LULC Class | Vegetation | Build Up and Bare Soil | Water Bodies | Total |
| Vegetation | 15 | 0 | 0 | 15 |
| Build Up and Bare Soil | 0 | 13 | 2 | 15 |
| Water Bodies | 1 | 1 | 13 | 15 |
| Total | 16 | 14 | 15 | 45 |
| | | Overal Accuracy | 87% | |
| | | Overal Kappa | 0.91 | |

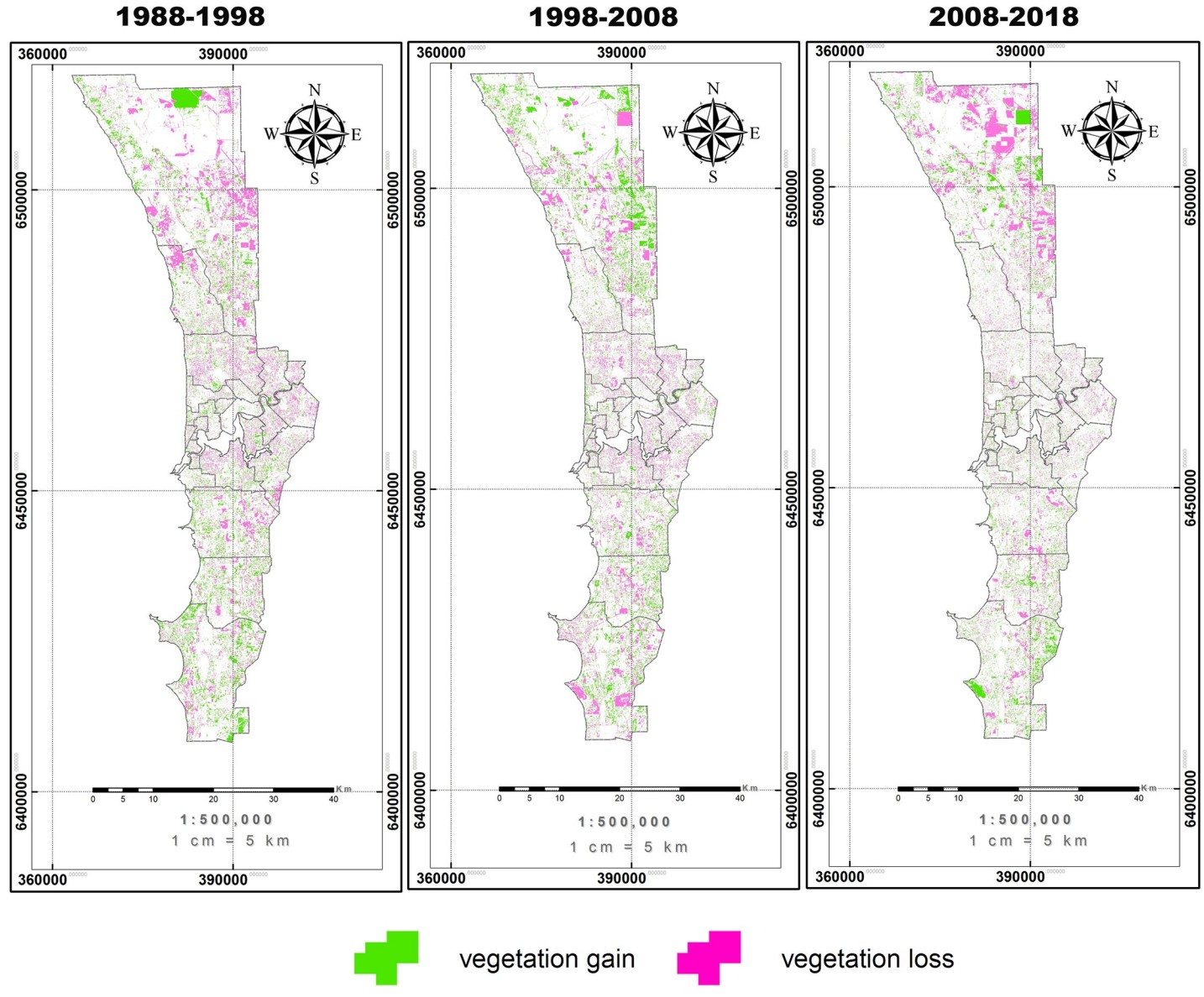

vegetation gain     vegetation loss

**Fig 5. Change in vegetation cover over the period 1988 to 2018 for the Perth region.** vegetation gain and vegetation loss are indicated.

**Table 4. Vegetation loss and gain (ha) between 1988 and 2018.**

| | | Period | | |
|---|---|---|---|---|
| | | 1988–1998 | 1998–2008 | 2008–2018 |
| Whole Area | Vegetation loss | 22,231 | 19,250 | 20,178 |
| | Vegetation gain | 15,538 | 15,838 | 12,591 |
| | Net loss | 6,692 | 3,412 | 7,587 |
| Golf courses | Vegetation loss | 68 | 27 | 40 |
| | Vegetation gain | 219 | 92 | 43 |
| | Net gain | 152 | 64 | 4 |

**Table 5. Results of Morphological Spatial Pattern Analysis (MSPA) analysis of connectivity of Perth's vegetation from 1988 to 2018.**

| Landscape type | Year | Perth Metropolitan Region | | Within golf courses | |
|---|---|---|---|---|---|
| | | Area (ha) | Proportion of total VCA (%) | Area (ha) | Proportion of total VCA (%) |
| Core | 1988 | 32,012 | 32.6 | 325 | 23.2 |
| | 1998 | 20,607 | 22.5 | 171 | 12.7 |
| | 2008 | 22,251 | 25.3 | 230 | 16.7 |
| | 2018 | 18,101 | 22.5 | 383 | 27.3 |
| Bridge | 1988 | 37,127 | 37.9 | 738 | 52.6 |
| | 1998 | 39,412 | 43.0 | 814 | 60.2 |
| | 2008 | 37,902 | 43.1 | 732 | 53.1 |
| | 2018 | 37,503 | 46.7 | 588 | 42.0 |
| Islet | 1988 | 10,396 | 10.6 | 31 | 2.2 |
| | 1998 | 16,436 | 17.9 | 78 | 5.8 |
| | 2008 | 12,922 | 14.7 | 15 | 1.1 |
| | 2018 | 12,349 | 15.4 | 37 | 2.7 |
| Perforation | 1988 | 1,099 | 1.1 | 3 | 0.2 |
| | 1998 | 486 | 0.5 | - | - |
| | 2008 | 834 | 1.0 | - | - |
| | 2018 | 406 | 0.5 | 6 | 0.4 |
| Edge | 1988 | 7,430 | 7.6 | 162 | 11.5 |
| | 1998 | 4,814 | 5.3 | 95 | 7.0 |
| | 2008 | 4,986 | 5.7 | 147 | 10.7 |
| | 2018 | 5,218 | 6.5 | 230 | 16.4 |
| Loop | 1988 | 5,670 | 5.8 | 112 | 8.0 |
| | 1998 | 6,295 | 6.9 | 146 | 10.8 |
| | 2008 | 5,607 | 6.4 | 209 | 15.2 |
| | 2018 | 3,669 | 4.6 | 129 | 9.2 |
| Branch | 1988 | 4,361 | 4.5 | 32 | 2.3 |
| | 1998 | 3,619 | 4.0 | 47 | 3.5 |
| | 2008 | 3,491 | 4.0 | 43 | 3.2 |
| | 2018 | 3,117 | 3.9 | 29 | 2.1 |

not contribute to connectivity in the landscape. The proportion of these classes increased through time from 24% in 1988 to 30% in 2008 and 2018.

Fig 6 shows that the core area was distributed mostly in the northern part of the city and their areas decreased significantly in later years. In 1988, the bridge class covered a large area of the city's central region but this decreased over time. In recent years, most of the vegetation cover in the central region of the city belongs to the islet, loop, edge, perforation and branch classes, illustrating that isolation became more serious in the central region of the city over the three decades.

The vegetation cover within golf courses also contributes to connectivity. The proportion of core area within golf courses fluctuated between 12% and 27%. Moreover, the largest proportion of vegetation in golf courses was classified as bridge but it experienced a downward trend from 52% to 41% in three decades. Of the remaining classes which do not contribute to connectivity, the edge and loop classes accounted for a higher proportion with each of them contributing 7% to 16% of total vegetation cover.

## 4 Discussion

We used Landsat imagery to characterize the patterns of vegetation change, habitat connectivity and the role of golf courses in maintaining green spaces and connectivity in an urban

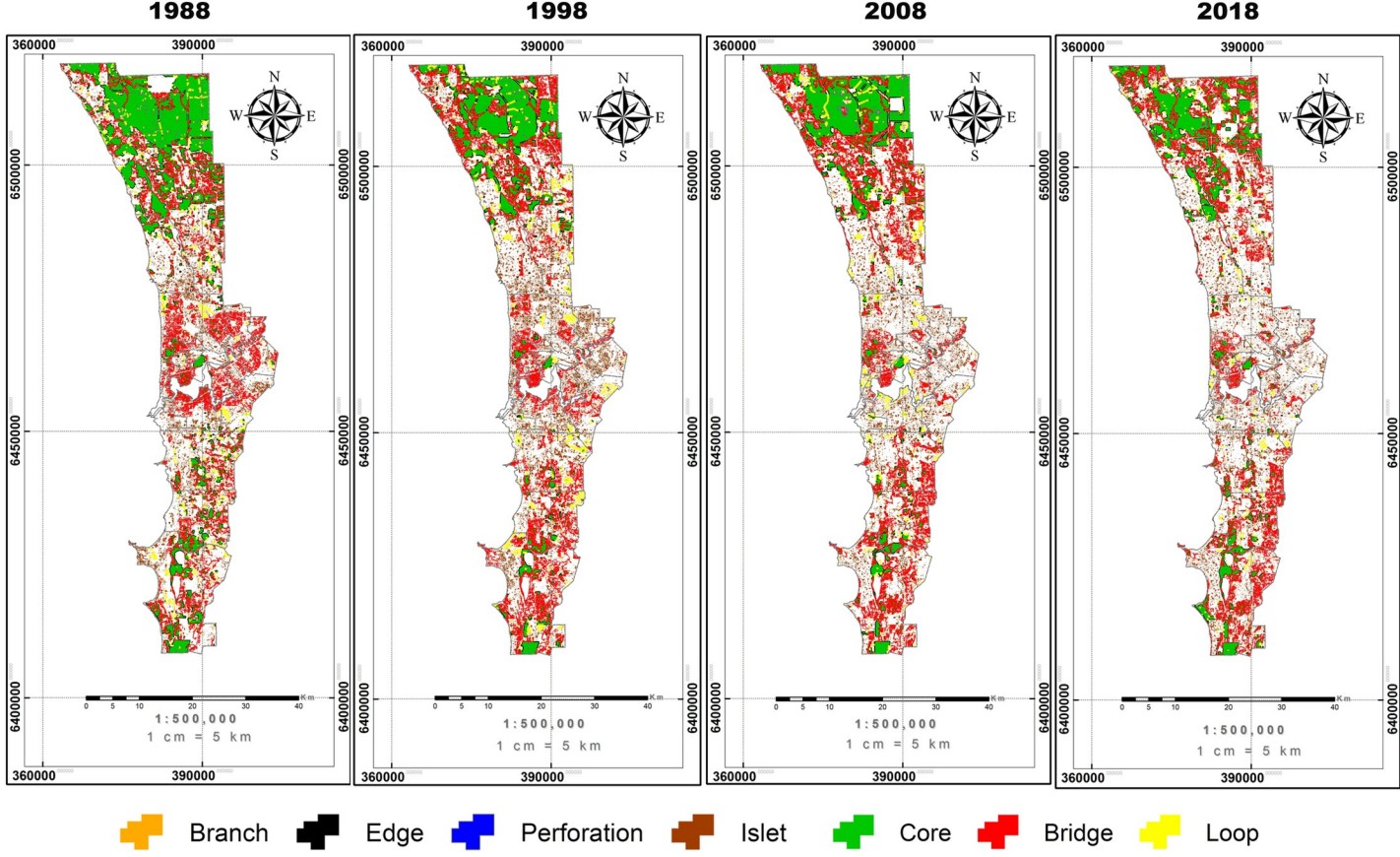

**Fig 6. Results of the Morphological Spatial Pattern Analysis (MSPA) for the Perth region from 1988 to 2018 at for four time steps.**

landscape. We found that deforestation led to a reduction in habitat connectivity in the Perth Metropolitan Region. However, golf courses can play an important role in maintaining vegetation and supporting biodiversity connectivity in urban landscapes. Previous studies have documented the increase in the urban footprint of Perth using multi-temporal urban expansion statistics derived from Satellite imagery [55]; However, their work did not address the issues of vegetation dynamics, nor the biodiversity and structure of the vegetation and habitat connectivity in the Perth Metropolitan Region. Therefore, our study addresses this gap.

## 4.1 Deforestation and urbanization

The vegetation dynamics in the major urban areas of Perth reflect the pattern of urbanization over time. The reduction in green space found in this study can be explained as a close relation to the process of development in this city. Over the last three decades, urban development in Perth has taken place at a fast rate [39]. In the early 1990s to 2006, Perth's population grew by around 1.8%, but the figure has nearly doubled since then [56]. Also, this study indicated that, from 1988 to 2018, Perth's urban footprint increased from 74,376 to 92,243 ha (Table 2) and is consistent with previous research in urban growth in this region [55]. In the last 20 years, on average, 740 ha/yr of urban and urban deferred zoned land was consumed by subdivision, and 830 ha/yr was consumed by construction in the Perth metropolitan area and nearby Peel region [57]. In addition to this expansion, the city has become denser with the construction of new residential dwellings in urbanised areas. Although vegetation gain occurred in some

places as a result of natural increase in canopy cover as urban vegetation grows through the conservation efforts, vegetation offset from development projects (e.g. mining), the plantation programs of the government taking place in bare soils in some suburbs, and efforts to increase green spaces from private land owners, the vegetation loss associated with urbanization has been more significant.

The difference in spatial patterns of vegetation loss are also related to the urban plans of this city. Our results show that between 1988 and 1998, there was significant vegetation loss in the central region of the city which can be linked to The Corridor Plan [58]. Historically, Perth's development pattern from 1970s to 1990s was based on linear corridors stretching out from the city's core, with large non-urban wedges between each of these corridors [58]. However, from 1998 to 2008 and from 2008 to 2018, deforestation was more significant in the outer sub-regions north-west and south-west of the city due to the adoption of Metroplan [59]. Perth recently has been divided into subregional areas, rather than corridors for planning purposes. The two coastal subregions (the North-West and South-West) included in our study have consistently achieved higher rates of population growth under Metroplan [60].

Originally the region was covered by woodlands dominated by eucalypts and banksias and coastal heath interspersed with chains of wetlands. Perth is home to a rich biodiversity, with more than 1,700 species of flowering plants and iconic species of threatened fauna in the region [61]. Therefore, such deforestation for urbanization has resulted in the devastating loss of significant natural habitats in this biodiversity hotspot city, leading to the designation as an endangered ecological community by the Australian Government [62].

## 4.2 Connectivity of green space networks in urban landscapes

The MSPA analysis indicates that the loss of vegetation as a consequence of urban expansion has led to a marked reduction in connectivity of green space networks in the Perth urban landscape. As only cores (the stepping stones between forest habitat patches) and bridges (the structural corridors to link core areas) can contribute to the connectivity between the habitat areas in the landscape [63], the reduction of these areas in Perth associated with urbanization throughout the time indicates the high impact of urban development on habitat connectivity. This analysis also shows the increase in proportion of islets, which are totally isolated patches, and other classes (perforations, loops, branches, edges) that cannot reach a new core habitat area for originating the potential movement [63]. Clearly, expansion of Perth city has fragmented the remaining blocks of natural habitat and increased isolation of natural habitats. This may reduce population and gene flow among patches and may disrupt the connection between subpopulations and a large regional population [64] and thus threaten the long-term viability of relict populations.

Fragmentation was obvious in the central region of the city and in recent decades it has become more serious in the outer parts of the city. This is critical as Perth is within a globally recognised biodiversity hotspot, which is home to rich biodiversity found nowhere else in the world. The connectivity in the major urban area of Perth is not only critical for the linkage of habitats within the Swan Coastal Plain but also for the connection of these coastal habitats to a large regional biosphere in south-western Australia.

The results also illustrated that a 'Core' area and a network of 'Bridge' types exist in the central and outer subregions of Perth. This is the consequence of early conservation efforts of the government which created protected areas such as Kings Park, Bold Park and other significant areas. Very few cities in the world have such large areas of natural bushland in the centre of a big city [65]. However, future urban growth will continue to put pressure on the biodiversity. If current policies (Perth and Peel@3.5million) are fully implemented, existing stocks of urban

and urban deferred land would be consumed by about 2075 [66]. The challenge for urban planning to preserve urban forests and biodiversity is thus increasing, and it is clear that planning for future expansion should also include large protected areas.

Our MSPA output with spatial distribution of seven classes provides fundamental information for future urban planning. There is a need to maintain important green spaces which are classified as cores and bridges in the city, especially in the central region where most of the natural vegetation exists as islands. Also, the MSPA branch classes can be used to identify candidate ecological restoration areas. The branch class can be thought of [67] as a foundation of a potential corridor that could, if revegetated, connect two spatially disjunct core areas to improve connectivity in the larger region.

Although other analyses, such as functional connectivity, should be taken into account in landscape connectivity assessment [30], the structural connectivity analysis in this study will be useful for determining the priority protection level and critical areas of the connecting corridor, informing conservation strategies at a variety of scales, especially when the biodiversity values of this region are suffering from various threats including deforestation, feral animals, weed incursions, more frequent fires through arson and tree disease [61].

### 4.3 The role of golf courses in maintaining urban forest

Our study indicates that golf courses account for a significant proportion of the urban area of Perth. This category of land use has been vital in maintaining green space in urban areas over the past thirty years. In contrast to the overall decline in urban green space, golf courses have preserved green spaces within urban settings and even created a net gain of vegetation cover over time. The highest net gain was seen in the period between 1988 to 1998 when some golf courses were established resulting in the planting of trees.

In the green 'matrix' of Perth, golf courses with their significant area of vegetation cover have contributed considerably to the connectivity in the urban landscape. A significant proportion of their green space was classified as core or bridge categories. The proportion of vegetation within golf courses classified as bridges was higher than in the whole study area. Golf courses with large areas of native vegetation provide "links" to other large natural patches of urban vegetation.

Although there are concerns with the environmentally negative impacts of golf courses as a source of pollution through pesticide and fertiliser usage [20], habitat modification [23] and high water usage [68], previous studies provide evidence about the biological values of golf courses, such as providing refugial habitat for urban-avoiding wildlife [22, 69–77]. Our study indicates that golf courses in urban settings have been maintaining large green space in urban settings and may play a role in biodiversity connectivity in the city.

### 4.4 Monitoring urban forest dynamics

In this study, we utilized medium-resolution satellite remote sensing data to identify land use classes, characterise vegetation dynamics and connectivity. The data maps the spatial and temporal patterns of land use types characterizing a consistent, detailed vegetation dynamic of the city [55, 78]. Clearly, the biophysical elements of urban landscapes are well-reflected through physical features (NDVI) derived from remote sensing data with an accuracy of up to 89%.

Despite these kinds of data, it is hard to describe the detailed information of ecosystems such as species composition and forest structure; to differentiate between the types of green space and habitat quality as the spatial resolution of the imagery is unable to differentiate between trees, shrubs, turf, groundcovers etc. However, medium-resolution satellite remote sensing data have and advantage in mapping land cover dynamics across large areas of big

cities over time when high resolution imagery is not available. In our study area, Landsat is the only platform that provides the opportunity to retrospectively assess vegetation trends over the last three decades. For monitoring urban forests in big cities, the large scale and long temporal datasets are more advantageous compared with datasets that focus only on discerning a specific land use type in a relatively small area [25, 79, 80]. This is because it allows spatially detailed identification of changes associated with development over time. Therefore, the approach described in this paper provides baseline information for sustainable urban planning and development. In addition, the MSPA analysis can further evaluate the dynamic of vegetation cover by the describing the spatial configuration of ecosystems at the pixel level, detecting changes of habitat connectivity over time [30].

## 5 Conclusions

With rapid urban expansion, the most meaningful question to address is how to balance urban development and urban forest preservation. Urbanization requiring deforestation is inevitable in many cities worldwide. Our study found a significant loss of vegetation cover in a biodiversity hotspot over three decades of urbanization, which led to a reduction in habitat connectivity in the urban landscape. A lesson learned from the experience of urbanization in Perth is that any future urban growth following the patterns observed over the past three decades will continue to put pressure on maintaining urban forest ecosystems and biodiversity conservation. As cities continue to grow in response to socio-economic development, considering all opportunities for urban biodiversity conservation is important. Urban conservation strategies must therefore consider not only the protected areas, but also the off-reserve sites.

Our study indicates that golf courses in urban settings have been maintaining green-area habitats and have played an important role in biodiversity connectivity in the city. Potentially, urban golf courses could become more purposefully managed for biodiversity conservation and the improvement of critical ecosystem services in urban areas. In rapidly urbanizing biodiversity hotspots like Perth, where fragmentation is one the biggest threats to biodiversity, the way that golf courses contribute to increase the connectivity in the intervening urban matrix should not be underestimated. Therefore, it is important for government authorities and golf courses owners to pay more attention in maintaining ecosystem health in urban golf courses.

## Acknowledgments

The authors thank Dr. Harry Eslick for his valuable advice in helping to design the research and Mr. Pham Dang Manh Hong Luan for his advice in data analysis.

## Author Contributions

**Conceptualization:** Thu Thi Nguyen, Richard Harper, Bernard Dell.

**Data curation:** Thu Thi Nguyen, Paul Barber.

**Formal analysis:** Thu Thi Nguyen, Tran Vu Khanh Linh.

**Methodology:** Thu Thi Nguyen, Tran Vu Khanh Linh.

**Supervision:** Paul Barber, Richard Harper, Bernard Dell.

**Validation:** Paul Barber.

**Writing – original draft:** Thu Thi Nguyen.

**Writing – review & editing:** Richard Harper, Bernard Dell.

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
