## [Decision Letter · Decision Letter 0]

19 Sep 2019

PONE-D-19-19510

Trends in vegetation dynamics associated with urban development: The role of golf courses

PLOS ONE

Dear Dr Thu T. Nguyen,

Thank you for submitting your manuscript to PLOS ONE. After careful consideration, we feel that it has merit but does not fully meet PLOS ONE’s publication criteria as it currently stands. Therefore, we invite you to submit a revised version of the manuscript that addresses the points raised during the review process.

Please note that there are minor changes needed to be done as all the reviewers identify minor revisions. At the beginning, I couldn't find reviewers for the review of your manuscript and I invited a number of reviewers. Four of them accepted to review and provided their comments.  We have looked over the comments from these reviewers and find that you should be able to readily accommodate these revisions.

We would appreciate receiving your revised manuscript by 18 October 2019. To enhance the reproducibility of your results, we recommend that if applicable you deposit your laboratory protocols in protocols.io, where a protocol can be assigned its own identifier (DOI) such that it can be cited independently in the future. For instructions see: http://journals.plos.org/plosone/s/submission-guidelines#loc-laboratory-protocols

We look forward to receiving your revised manuscript.

Kind regards,

Eda Ustaoglu, PhD

Academic Editor

PLOS ONE

Journal Requirements:

2. We note that Figure 1 in your submission contain map/satellite images which may be copyrighted. All PLOS content is published under the Creative Commons Attribution License (CC BY 4.0), which means that the manuscript, images, and Supporting Information files will be freely available online, and any third party is permitted to access, download, copy, distribute, and use these materials in any way, even commercially, with proper attribution. For these reasons, we cannot publish previously copyrighted maps or satellite images created using proprietary data, such as Google software (Google Maps, Street View, and Earth). For more information, see our copyright guidelines: http://journals.plos.org/plosone/s/licenses-and-copyright.

We note that one or more of the authors are employed by a commercial company: ArborCarbon Pty Ltd

Reviewers' comments:

Reviewer's Responses to Questions

**Comments to the Author**

1. Is the manuscript technically sound, and do the data support the conclusions?

Reviewer #1: Partly

Reviewer #2: Yes

Reviewer #3: Yes

Reviewer #4: Yes

2. Has the statistical analysis been performed appropriately and rigorously? 

Reviewer #1: N/A

Reviewer #2: Yes

Reviewer #3: Yes

Reviewer #4: Yes

3. Have the authors made all data underlying the findings in their manuscript fully available?

Reviewer #1: Yes

Reviewer #2: Yes

Reviewer #3: Yes

Reviewer #4: Yes

4. Is the manuscript presented in an intelligible fashion and written in standard English?

Reviewer #1: Yes

Reviewer #2: Yes

Reviewer #3: Yes

Reviewer #4: Yes

5. Review Comments to the Author

Reviewer #1: A great manuscript with an interesting topic.

And while I am very supportive of publishing this work, there were few issues that I would like the authors to address first, most of them are technical and few are in need of clarifications:

1. I am not too sure about your last statement in the Abstract that suggest "we need to consider before clearing for urban development". An already developed area becomes a fair game in my opinion and you may need to qualify that statement.

2. The general idea of large urban environments being non conductive to healthy lifestyles is generally well founded but maybe beyond the amount of green?

While there are few publications about these topics, most of them tend to be rather tenuous at best.

3. REF 29 is not correct? Look for Tucker, Huete, Myneni, Prince, for an NDVI reference

4. Quality all Figures is rather very poor

5. Page #5, there was hardly any useful information on how the Atm. Correction was carried. You need to clarify this further so readers can replicate and or understand better.

5. NDVI is higher usually due to "Red" being low which in turn is due consumption by chlorophyll during photosynthesis.

6. There are issues with the classification based on images from the month of December only (Summer). There was little in the way of what exact dates, what data, how did you remove clouds, etc... This is critical since it will impact the change detection which is based on NDVI, which is very sensitive to noise in the data and the time the data came from? You will need to elaborate on your exact methods and data analysis here.

6. The concept of vegetation cover gain in urbanized areas is a bit of a misnomer, as how would that happen? is it urban abandonment? or other mechanism? Please clarify.

7. You need to clarify the meaning and exact definitions of the proposed MSPA metrics? What does COR/Bridge, etc.. measure here? I realize you pointed and referenced two publications but your work need to be stand alone to a certain degree.

8. I also find the reference to " Golf courses" as a fully functional and reliable habitats a bit confusing considering their high traffic and human impacts? Please clarify.

Generally the manuscript is well written and easy to read and understand, but it lacked critical technical information and clarift in some key areas (Data analysis, Habitat analysis). I suggest you address these shortcoming.

I did catch few issues in the text that are enclosed in the form of comments in the PDF file.

Reviewer #2: This study do not intend to explain the 'dynamic'- nets effect of many factors such as climate, abiotic environment, biotic interaction, disturbance history etc. In vegetation cover over Perth metropolitan, but seek to describe 'only' the spatio-temporal land use change pattern associated with urbanization in the area. Hence, the title should be reframe to capture only this aspect.

Reviewer #3: kindly improve the map resolution of all figures

the accuracy assessment of the classification is not exit the manuscript, it is needed to be added.

kindly add the lat/long, scale, and north arrow on the border of layout.

Reviewer #4: Golf courses have potential in habitat but to think they are superior in habitat protection than natural areas is the problem I have.

I also think you kept moving forward and back on the focus of the study. Please refer to my comments attached.

6. PLOS authors have the option to publish the peer review history of their article (what does this mean?). If published, this will include your full peer review and any attached files.

Reviewer #1: No

Reviewer #2: No

Reviewer #3: Yes: Dr. Mohamed A.E. AbdelRahman

Reviewer #4: Yes: ADAMS OSMAN

---

## [Author Response · Author response to Decision Letter 0]

21 Nov 2019

Reviewer 1: We have incorporated all of your comments into to our revision. They were very useful.

Reviewer 2: We have incorporated all of your comments into to our revision. Thank you for your helpful suggestions.

Reviewer 3: We have incorporated all of your comments into to our revision. Thank you for your help.

Reviewer 4: We have incorporated all of your comments into to our revision. They were very helpful.

---

## [Decision Letter · Decision Letter 1]

11 Dec 2019

PONE-D-19-19510R1

Vegetation trends associated with urban development: the role of golf courses

PLOS ONE

Dear Dr Nguyen,

Thank you for submitting your manuscript to PLOS ONE. After careful consideration, we feel that it has merit but does not fully meet PLOS ONE’s publication criteria as it currently stands. Therefore, we invite you to submit a revised version of the manuscript that addresses the points raised during the review process.

At this stage, Reviewer 1 proposes some changes, though they are minor to be addressed in the revised version of the manuscript.

We would appreciate receiving your revised manuscript by January 10, 2020. To enhance the reproducibility of your results, we recommend that if applicable you deposit your laboratory protocols in protocols.io, where a protocol can be assigned its own identifier (DOI) such that it can be cited independently in the future. For instructions see: http://journals.plos.org/plosone/s/submission-guidelines#loc-laboratory-protocols

We look forward to receiving your revised manuscript.

Kind regards,

Eda Ustaoglu, PhD

Academic Editor

PLOS ONE

Reviewers' comments:

Reviewer's Responses to Questions

**Comments to the Author**

1. If the authors have adequately addressed your comments raised in a previous round of review and you feel that this manuscript is now acceptable for publication, you may indicate that here to bypass the “Comments to the Author” section, enter your conflict of interest statement in the “Confidential to Editor” section, and submit your "Accept" recommendation.

Reviewer #1: All comments have been addressed

Reviewer #2: All comments have been addressed

2. Is the manuscript technically sound, and do the data support the conclusions?

Reviewer #1: Yes

Reviewer #2: Yes

3. Has the statistical analysis been performed appropriately and rigorously? 

Reviewer #1: Yes

Reviewer #2: Yes

4. Have the authors made all data underlying the findings in their manuscript fully available?

Reviewer #1: Yes

Reviewer #2: Yes

5. Is the manuscript presented in an intelligible fashion and written in standard English?

Reviewer #1: Yes

Reviewer #2: Yes

6. Review Comments to the Author

Reviewer #1: The authors did a great job at addressing most reviewers comments, including mine.

There are few additional and minor changes that I noted on the enclosed PDF file and would like the authors to address.

Few do require serious changes especially the quality of the plots, figures, and captions, some require editing to clarify assumptions and some require rewording to make it clear that for example golf courses green, while is green and will be picked up by NDVI, is not the same quality green as natural and from a habitat perspective.

I did note few more comments on the attached PDF.

Reviewer #2: (No Response)

7. PLOS authors have the option to publish the peer review history of their article (what does this mean?). If published, this will include your full peer review and any attached files.

Reviewer #1: No

Reviewer #2: Yes: Tosin Sunday Adeyemi

---

## [Author Response · Author response to Decision Letter 1]

6 Jan 2020

Reviewer 1: We have incorporated all of your comments into our revision. They were very helpful.

---

## [Editor Report · Decision Letter 2]

8 Jan 2020

Vegetation trends associated with urban development: the role of golf courses

PONE-D-19-19510R2

Dear Dr. Nguyen,

We are pleased to inform you that your manuscript has been judged scientifically suitable for publication and will be formally accepted for publication once it complies with all outstanding technical requirements.

With kind regards,

Eda Ustaoglu, PhD

Academic Editor

PLOS ONE
---

## [Editor Report · Acceptance letter]

13 Jan 2020

PONE-D-19-19510R2 

Vegetation trends associated with urban development: the role of golf courses 

Dear Dr. Nguyen:

I am pleased to inform you that your manuscript has been deemed suitable for publication in PLOS ONE. Congratulations! Your manuscript is now with our production department. 

With kind regards,

on behalf of

Dr. Eda Ustaoglu 

Academic Editor

PLOS ONE